# LogicDP: Creating Labels for Graph Data via Inductive Logic Programming

**Yuan Yang[1], Faramarz Fekri[1], James C. Kerce[1] & Ali Payani[2]**
Georgia Institute of Technology[1], Cisco[2]
`{yyang754@,faramarz.fekri@ece.,clayton.kerce@gtri.}gatech.edu`
`apayani@cisco.com`

## Abstract

Graph data, such as scene graphs and knowledge graphs, see wide use in AI systems. In real-world and large applications graph data are usually incomplete, motivating graph reasoning models for missing-fact or missing-relationship inference. While these models can achieve state-of-the-art performance, they require a large amount of training data.

Recent years have witnessed the rising interest in label creation with data programming (DP) methods, which aim to generate training labels from heuristic labeling functions. However, existing methods typically focus on unstructured data and are not optimized for graphs. In this work, we propose LogicDP, a data programming framework for graph data. Unlike existing DP methods, (1) LogicDP utilizes the inductive logic programming (ILP) technique and automatically discovers the labeling functions from the graph data; (2) LogicDP employs a budget-aware framework to iteratively refine the functions by querying an oracle, which significantly reduces the human efforts in function creations. Experiments show that LogicDP achieves better data efficiency in both scene graph and knowledge graph reasoning tasks.

## 1 Introduction

Graph data are widely used in many applications as structured representations for complex information. In the visual domain, a *scene graph* can be used to represent the semantic information of an image. As shown in Figure 1a, each node in the graph corresponds to a localized entity (e.g., $x_1$, $x_2$, and $x_3$) and each edge represents the semantic relations between a pair of entities (e.g., $\langle x_3, \texttt{Has}, x_1 \rangle$). Similarly, knowledge graphs (KGs) represent real-world facts with entities and the relations that connect them. Standard KGs such as Freebase Toutanova & Chen (2015) and WordNet (Bordes et al., 2013) consist of facts that describe commonsense knowledge and have played important roles in many applications (Yang et al., 2017; Sun et al., 2019; Mitchell et al., 2018; Yang et al., 2022).

Graph datasets are usually incomplete and new facts can be inferred from the existing facts. For example, in Figure 1a, the class label of $x_3$ is missing, and one can infer that it is a `Car` because $x_3$ `Has` a `Wheel` and a `Window`. This process is referred to as *graph reasoning*. A large body of research has been proposed to address this task. For scene graph reasoning, methods such as iterative message passing (Xu et al., 2017), LSTM (Zellers et al., 2018) and GNN (Yang et al., 2018) are proposed. And for KG reasoning, a variety of graph embedding methods (Bordes et al., 2013; Sun et al., 2019) are proposed. These graph reasoning methods rely on standard supervised training, where the model is fed with a fixed set of data that are curated beforehand. Such training leads to a state-of-the-art performance when a large amount of data is available. However, this approach is shown to be suboptimal with respect to data efficiency (Misra et al., 2018; Shen et al., 2019), and creating a sufficiently large training dataset manually can be expensive.

In this work, we approach this problem by adopting the *data programming* (DP) paradigm (Ratner et al., 2016), which aims at creating a large high-quality labeled dataset from a set of *labeling functions* that generate noisy, incomplete, and conflicting labels. To this end, we propose LogicDP, a DP framework that creates training data for graph reasoning models. Compared to the existing domain-general DP frameworks (Ratner et al., 2016; 2017), LogicDP utilizes the inductive logic

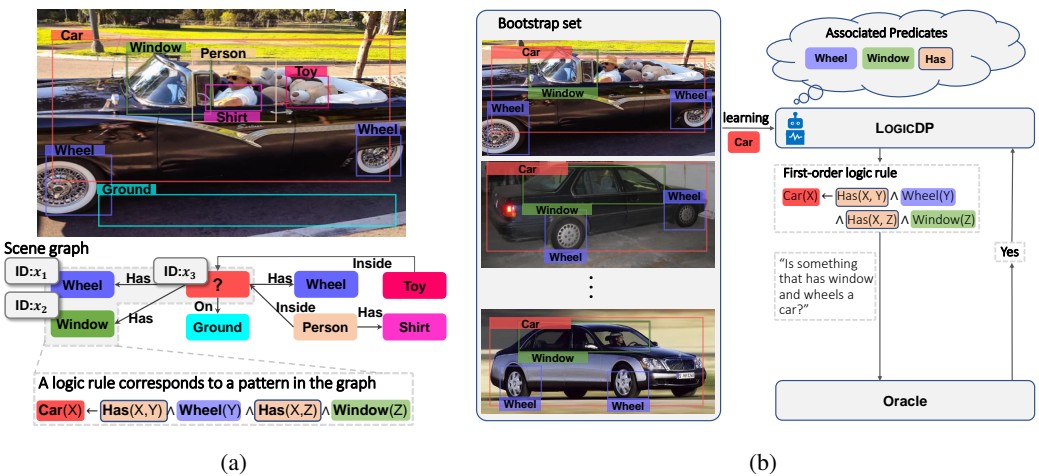

(a)   (b)

Figure 1: (a) An image can be represented by a scene graph and a logic rule can be used to represent the shaded pattern in the graph. (b) Learning and refining a labeling function for the unary predicate `Car`.

programming technique and can automatically generate labeling functions in the form of first-order logic rules from a small labeled set; LOGICDP also employs a budget-aware framework that refines the functions through weak supervision from an oracle.

The LOGICDP framework is flexible and agnostic to the choice of the graph reasoning model and the ILP method. In experiments, we evaluate LOGICDP on scene graph and KG datasets together with several strong generic weakly-supervised methods, which suggests that LOGICDP scales well with large graph datasets and generalizes better than the baselines. We also showcase the training with human oracles and discuss its potential as a novel human-in-the-loop learning paradigm.

## 2   RELATED WORK

**Data Programming**. LOGICDP is related to the recent advances in data-centric AI and data programming (Ratner et al., 2016; 2017; Varma & Ré, 2018), which aims at a new paradigm for training and evaluating ML models with weak supervision. Existing frameworks such as Snorkel (Ratner et al., 2017) show great potential in this direction but are also limited in applications where expert labeling functions are difficult or expensive to construct, and they do not utilize the rich semantic information in graphs to automate the process. In LOGICDP, we incorporate the ILP technique and automatically generate the labeling functions from a small set of graph data.

**Graph reasoning and inductive logic programming**. Graph reasoning is the fundamental task performed on graph data. It can be addressed by many approaches. For example, graph embedding methods (Bordes et al., 2013; Sun et al., 2019), GNN-based methods (Yang et al., 2018; Zhang et al., 2020) and various deep models (Xu et al., 2017; Zellers et al., 2018). While these data-driven models achieve state-of-the-art performance, they require many samples to train. On the other hand, graph reasoning can be solved by finding a multi-hop path in the graph that predicts the missing facts (Guu et al., 2015; Lao & Cohen, 2010; Lin et al., 2015; Gardner & Mitchell, 2015; Das et al., 2016). Specifically, *inductive logic programming* (ILP) methods (Galárraga et al., 2015; Evans & Grefenstette, 2018; Payani & Fekri, 2019; Campero et al., 2018; Yang & Song, 2020; Yang et al., 2017) learns to predict missing facts by searching for such paths and representing them as logic rules. Compared to the previous methods, ILP methods are more data-efficient and offer better interpretability. In this work, we investigate using logic rules learned by ILP methods as labeling functions and generating triples to train a data-driven graph reasoning model.

## 3   PRELIMINARIES AND PROBLEM STATEMENT

We consider graphs such as scene graphs and KGs that consist of a set of *facts* in the form of head-predicate-tail *triples*. Here, we use the scene graph in Figure 1a as a running example. Formally, we

define such a graph as $\mathcal{G} = \langle \mathcal{X}, \mathcal{P}, \mathcal{T} \rangle$, where $\mathcal{X}$ denotes the set of entities in the graph and in the case of Figure 1a, $\mathcal{X} = \{x_1, x_2, x_3, ...\}$; $\mathcal{P}$ denotes the set of predicates and example predicates $P \in \mathcal{P}$ in Figure 1a are `Has`, `Inside`, `Wheel` and `Car`; finally, $\mathcal{T}$ denotes the set of triples in this graph, where each triple $\boldsymbol{t}$ consists of two entities and a predicate $\mathcal{T} = \{\boldsymbol{t} := \langle x, P, x' \rangle | x, x' \in \mathcal{X}, P \in \mathcal{P}\}$. There are two types of predicates: (1) *binary* predicates are commonly used in KGs, which correspond to relations between two entities. For example in Figure 1a, `Has` is a binary predicate and triples involving it are $\{\langle x_3, \texttt{Has}, x_1 \rangle, \langle x_3, \texttt{Has}, x_2 \rangle, ...\}$; they effectively represent the edges in $\mathcal{G}$. On the other hand, (2) *unary* predicates are not typically used in KGs but are common in scene graphs, which correspond to the class label or attributes of a single entity. In Figure 1a, this can be `Wheel` and `Window`, and triples of this type are $\{\langle x_1, \texttt{Wheel}, x_1 \rangle, \langle x_2, \texttt{Window}, x_2 \rangle, ...\}$ (we duplicate the same entity for unary predicate for notation consistency); they effectively represent the class labels of the entities.

**Graph reasoning**. The fundamental task of graph reasoning is to infer missing facts from the existing ones. These missing facts are referred to as *queries* and take the form of $\langle x, ?, x' \rangle$. In Figure 1a, a query can be $\langle x_3, ?, x_3 \rangle$ which asks for the missing class labels of $x_3$. In this work, we are interested in approaching this task by training a graph reasoning model. Formally, let $f_\theta(\boldsymbol{t})$ be a graph reasoning model parameterized by $\theta$, which infers the score of how likely a triple $\boldsymbol{t}$ exist in the graph, and let $\mathcal{T}$ be a set of training triples, one trains the model by minimizing the following margin-based objective

$$\mathcal{L}(\theta; \mathcal{T}) = \frac{1}{N} \sum_{\boldsymbol{t} \in \mathcal{T}} \sum_{\boldsymbol{t}' \in \mathcal{N}(\boldsymbol{t})} 1 - f_\theta(\boldsymbol{t}) + f_\theta(\boldsymbol{t}'), \tag{1}$$

where $\mathcal{N}(\boldsymbol{t})$ denotes the random triples generated by corrupting the entities $x, x'$ in the triple $\langle x, P, x' \rangle$. Note that this is different from classical objectives such as cross-entropy and is widely used in graph reasoning tasks because there are usually no explicit negative triples in the graph (Refer to details in (Guu et al., 2015; Bordes et al., 2013)). Our goal is to generate enough triples such that the model $f_\theta$ is sufficiently trained via Eq.(1).

**Data programming for graph reasoning**. Acquiring a large number of training triples manually is expensive. We adopt the data programming (DP) paradigm and propose to generate triples from a set of *labeling functions*. Similar to the reasoning model $f_\theta$, these functions infer the predicate given a query, but they are noisy and likely to conflict with each other and one needs to aggregate the outputs to generate the triples. Additionally, existing DP methods (Ratner et al., 2017; 2016) rely on domain experts to manually create good labeling functions. We observe that graph data contain rich semantic information and propose to automatically learn the labeling functions from the graph. Once the functions are learned from the data, one also needs a principled framework to incorporate human weak supervision (e.g., expert confidence scores on the functions) and refine the functions.

In summary, we specify the three challenges that need to be addressed to generate high-quality training triples for graph reasoning: (1) **Function Generation**: how to automatically generate labeling functions? (2) **Function refinement**: how to incorporate human weak supervision and refine the labeling functions? (3) **Label aggregation**: how to aggregate the labeling function outputs into the training triples usable for Eq.(1)? We address these challenges section 4.

# 4 PROPOSED METHOD

## 4.1 FUNCTION GENERATION

The first challenge we address is to automatically learn labeling functions from graphs. We assume a small *bootstrap* set $\mathcal{T}_{init}$ is given, from which we will generalize the functions. This is a realistic setting because many graph datasets such as Visual Genome (VG) (Krishna et al., 2016) have incomplete graphs with sparse connections (i.e., facts), while the data is insufficient for supervised training for $f_\theta$ (especially for the rare predicates), one can use them to construct functions with good heuristics. For example, in Figure 1b, a few scene graphs associated with predicates `Car`, `Wheel`, etc. are given as the bootstrap set, and we can generalize a heuristic that "Anything that has a `Wheel` and `Window` is a `Car`", which can be later used as a function to generate triples for `Car`.

To generalize functions from the bootstrap set $\mathcal{T}_{init}$, we need **(T1)** a function representation that is interpretable as they need to be refined by humans later in the process, and **(T2)** a data-efficient approach to generate functions because $\mathcal{T}_{init}$ is small.

**Logic rules as labeling functions**. The first-order logic rule severs as a principled representation for **(T1)**. One common logic rule family used for graphs is the chain-like Horn rules:

$$R : P(x, x') \leftarrow P^{(1)}(x, z_1) \wedge P^{(2)}(z_1, z_2)... \wedge P^{(T)}(z_{T-1}, x'). \tag{2}$$

A rule $R$ of this family consists of two parts: the head, which is $P(x, x')$, and the body which is $P^{(1)}(x, z_1) \wedge P^{(2)}(z_1, z_2)... \wedge P^{(T)}(z_{T-1}, x')$. In first-order logic, a binary predicate $P$ can be treated as a function $P : \mathcal{X} \times \mathcal{X} \mapsto \{0, 1\}$. In other words, it predicts `True` or `False` given a pair of entities (and similarly for unary predicates). The rule body is the conjunction (i.e., logical `And`) of a set of predicate functions, and it is `True` if all of them are `True`. Therefore, the logic rule $R$ is effectively a binary labeling function $R : \mathcal{X} \times \mathcal{X} \mapsto \{0, 1\}$, which encodes the statement that "$P(x, x')$ is `True` if the body $P^{(1)}(x, z_1) \wedge ... \wedge P^{(T)}(z_{T-1}, x')$ is `True`". Note that one can express other graph patterns with more complex rules. For example, the logic rule in Figure 1a corresponds to a tree consisting of two paths. For notation simplicity, we focus on the chain-like rules for the remaining contents and one can extend it to complex rules with different ILP methods.

**Rule learning via Inductive logic programming**. A chain-like rule $R$ can be learned from the bootstrap set $\mathcal{T}_{init}$ efficiently via inductive logic programming (ILP). Many ILP-based methods formalize this problem as a search problem, where given a query $\langle x, ?, x' \rangle$, one searches for a path in the graph from $x$ to $x'$ that entails the missing predicate. In particular, a chain-like rule of the form Eq.(2) corresponds to a path $x \xrightarrow{P^{(1)}} ... \xrightarrow{P^{(T)}} x'$ in the graph. Therefore, learning the logic rule is equivalent to finding the path from a graph. In general, learning explicit paths in the graph requires fewer data and ILP methods are significantly more data-efficient (Yang & Song, 2020).

**(T2)** is a self-contained ILP problem: given a graph with bootstrap triples $\mathcal{G} = \{\mathcal{X}, \mathcal{P}, \mathcal{T}_{init}\}$, one learns a set of logic rules for each $P \in \mathcal{P}$. This can be solved with any off-the-shelf ILP methods. Here, we briefly introduce the underlying methodology and leave the details in Appendix A.

In LOGICDP, we follow the differentiable backward-chaining ILP formalism, which learns the rule by training a model that searches for the optimal path in the graph. This is done by *performing a graph random walk with soft attention*. Let $\boldsymbol{M} \in \{0, 1\}^{|\mathcal{X}| \times |\mathcal{X}|}$ be the adjacency matrix of a graph. For a graph $\mathcal{G}$ with $|\mathcal{P}| = K$, there are $K$ matrices $\mathcal{M} = \{\boldsymbol{M}_1, ..., \boldsymbol{M}_K\}$. Let $v_x, v'_x$ be the one-hot vectors of entity $x, x'$; suppose we choose a path $x \xrightarrow{P^{(1)}} ... \xrightarrow{P^{(T)}} x'$, we can compute the score of how likely the path exists for $x, x'$ as

$$v^{(T)} = v_x^\top \prod_{t=1}^{T} \boldsymbol{M}^{(t)}, \qquad \text{score}(v_x, v'_x) = v^{(T)} \cdot v'_x, \tag{3}$$

where $\boldsymbol{M}^{(t)} \in \mathcal{M}$ is the adjacency matrix of the predicate used in $(t)$th step. One can verify that the $j$th element $v_j^{(T)}$ is the count of unique paths from $x$ to $j$ (Guu et al., 2015). Therefore, searching for the chain-like rule is equivalent to searching for a sequence of matrix multiplications $\boldsymbol{M}^{(t)}, t = 1, ..., T$. This hard-search problem can be relaxed into learning the weighted sums of all possible paths

$$\text{score}(v_x, v'_x | \alpha, \beta) = v_x^\top \sum_{t'=1}^{T} \alpha^{(t')} \left( \prod_{t=1}^{t'} \sum_{k=1}^{K} \beta_k^{(t)} \boldsymbol{M}_k \cdot v'_x \right), \tag{4}$$

where $\alpha = [\alpha^{(1)}, ..., \alpha^{(T)}]^\top$ is the path attention vector, and $\beta^{(t)} = [\beta_1^{(t)}, .., \beta_K^{(t)}]^\top$ is the matrix attention vector at $(t)$th step; they are generated from differentiable models such as RNN and transformer (Yang et al., 2017; Yang & Song, 2020).

Finally, since the model is differentiable via attention, one can train the model to maximize Eq.(4) for triples in $\mathcal{T}_{init}$ and extract the rules by sampling from the attention multinomials. We leave the training details in Appendix A because, as mentioned above, LOGICDP is agnostic to the ILP model and it is a self-contained task. For the remaining contents, we will assume the training is complete and a set of logic rules $\mathcal{R}_{cand}$ is generated for each predicate $P \in \mathcal{P}$.

### 4.2 FUNCTION REFINEMENT

We have now obtained a set of logic rules in the form of Eq.(2) via ILP, which generalizes predictive heuristics (e.g., Figure 1b) from a small bootstrap set $\mathcal{T}_{cand}$. However, these rules are potentially noisy

---

**Algorithm 1:** Training graph reasoning model with LOGICDP

---

**Input:** Graph $\mathcal{G}$, bootstrap set $\mathcal{T}_{init}$, budget $B$, threshold $\gamma$
**Init:** Reasoning model $f_\theta$, belief set $\mathcal{T} = \mathcal{T}_{init}$, refined rules $\mathcal{R} = \{\}$
$\mathcal{R}_{cand} \leftarrow$ Apply ILP to $\mathcal{G}$
**for** $i = 1, ..., B$ **do**
    $R \leftarrow \arg\max_{R \in \mathcal{R}_{cand}} cov(R), s.t.\ supp(R) > \gamma$
    $\lambda \leftarrow$ Obtain oracle score for $R$
    $\mathcal{R} \leftarrow \mathcal{R} \cup \{\langle \lambda, R \rangle\}$ ; $\mathcal{R}_{cand} \leftarrow \mathcal{R}_{cand}/\{R\}$
    $\mathbf{M}_{\text{label}}(\boldsymbol{t}|\mathcal{R}) \leftarrow$ Construct labeling model with Eq.(7)
    $\mathcal{T} \leftarrow$ Update belief set with Eq.(5)
    **repeat**
        Sample triples from $\mathcal{T}$
        Train $f_\theta$ by updating $\theta$ with Eq.(1)
    **until** *Convergence*;
**end**

---

and subject to spurious correlations in the data, which are harmful to generating labels. Therefore, one must refine the rules via human weak supervision. Note that this process can be very expensive in some data programming frameworks (Ratner et al., 2017) because the expert needs to manually create the functions, whereas in LOGICDP, the expert only needs to refine them.

**Budget-aware function refinement**. An effective refinement process should minimize human effort. We propose a budget-aware framework that iteratively refines the rule by querying an oracle and obtaining a confidence score. Let $B$ be the budget (the maximum number of interactions with the oracle) and $\mathcal{R}_{cand}$ be the initial set of rules obtained from ILP for predicate $P$ (one can extend it to sets of all predicates readily). The goal is to obtain a refined rule subset $\mathcal{R} = \{\langle \lambda_j, R_j \rangle | R_j \in \mathcal{R}_{cand}\}_{j=1}^{B}$ with $\lambda_j \in [0, 1]$ be the oracle assigned confidence score for rule $R_j$.

With a limited budget $B$, one should pick rules that balance between **coverage**: the number of triples that can be generated from the rule, and **support**: the number of generated triples that are correctly labeled. We quantify them through the *belief set*: the set of triples generated by the refined rules during the refinement process. Formally, we denote $\mathcal{T}_i$ as the belief set at $i$-th iteration and we have

$$\mathcal{T}_0 = \mathcal{T}_{init}, \quad \mathcal{T}_i = \mathcal{T}_{init} \cup \{\langle x, P^*, x' \rangle | P^* = \arg\max_{P \in \mathcal{P}} \mathbf{M}_{\text{label}}(\langle x, P, x' \rangle | \mathcal{R}), \langle x, x' \rangle \in \mathcal{X}^2\}, \quad (5)$$

where $\mathbf{M}_{\text{label}}$ (introduced in section 4.3) is the model that generates aggregated score of triple $\langle x, P, x' \rangle$ from $\mathcal{R}$, and $P^*$ is the inferred label with the largest score. In other words, $\mathcal{T}_i$ is the set of triples outputted by the labeling model at $i$-th iteration. It is first initialized into the bootstrap set $\mathcal{T}_{init}$; then for every iteration $i = 1...B$, we first update the labeling model $\mathbf{M}_{\text{label}}$, and then reconstruct the belief set by inferring the labels for all pairs of entities and combining the inferred triples with $\mathcal{T}_{init}$.

Now we define the support and coverage of rule $R$. Let $\mathcal{T}^R = \{\langle x, P, x' \rangle | R(x, x') = 1, \langle x, x' \rangle \in \mathcal{X}^2\}$ be the set of triples generated by rule $R$. Then, we define the *true positive* subset of $\mathcal{T}^R$ at $i$-th iteration as $\mathcal{T}_i^{R,tp} = \mathcal{T}^R \cap \mathcal{T}_i$, which is the intersection of the generated triples and the belief set. Finally, we compute the support and coverage as

$$supp(R) = |\mathcal{T}_i^{R,tp}|/|\mathcal{T}^R| \qquad\qquad cov(R) = |\mathcal{T}^R|/|\mathcal{X}|^2. \qquad (6)$$

In other words, $supp(R)$ is the ratio of the true positives in $\mathcal{T}^R$ and $cov(R)$ is the size of $\mathcal{T}^R$ divided by the total number of possible entity pairs.

For each iteration, we measure the support and coverage for each rule in $\mathcal{R}_{cand}$ and send the best one to the oracle for weak supervision (i.e., $\lambda$). This is done by (1) filtering out rules with support below a threshold $\gamma$; and (2) choosing the rule with the largest coverage and sending it to the oracle (Algorithm 1). In the experiments, we demonstrate the noise introduced by the incorrect labels can be alleviated by the aggregation model introduced in section 4.3. A similar strategy is also used in previous work (Ratner et al., 2017; Varma & Ré, 2018).

### 4.3 LABEL AGGREGATION

Once we obtain the refined rule set $\mathcal{R}$, the last task is to aggregate the noisy and conflicting labels generated by different rules and produce the final labels for training the reasoning model $f$. In previous work (Ratner et al., 2017), this process is formalized with a generative model, which can be expensive for graph data as it involves probabilistic inference. In LOGICDP, we approach this task by utilizing the *posterior regularization* (PR) (Ganchev et al., 2010) technique as it is a principled framework that can efficiently incorporate weak supervision and previous work (Guo et al., 2018; Hu et al., 2016) has shown it is effective for alleviating the noise introduced by the imperfect logic rules.

Formally, let $\mathcal{R} = \{\langle \lambda_j, R_j \rangle\}_{j=1}^{B}$ be the set of refined rules and confidence scores of predicate $P$ (one can extend it to sets of all predicates readily). The PR technique treats $\mathcal{R}$ not as independent functions but as a set of constraints imposed on the posterior of the reasoning model $f_\theta$. This leads to a constrained optimization problem (Appendix B). Solving the problem yields a labeling model

$$\mathbf{M}_{\text{label}}\left(\boldsymbol{t}|\mathcal{R}\right) \propto f_\theta\left(\boldsymbol{t}\right) \exp\left(-C \sum_{j=1}^{B} \lambda_j(1 - R_j(x, x'))\right), \quad \boldsymbol{t} := \langle x, P, x' \rangle, \tag{7}$$

where $C$ is the weight of the exponential term. The score of $\mathbf{M}_{\text{label}}$ is proportional to the product of $f_\theta(\boldsymbol{t})$ and an exponential term. This term is effectively a log-linear model that sums over the scores of rules in $\mathcal{R}$ weighted by $\lambda$s. $\mathbf{M}_{\text{label}}$ balances between the supervision from $\mathcal{R}$ and the inference from $f_\theta$: with a large $C$, the model generates scores that are close to that from $\mathcal{R}$ and vice versa.

With $\mathbf{M}_{\text{label}}$, we train the reasoning model $f_\theta$ by constructing and sampling from the belief set with Eq.(5). We summarize the overall pipeline in Algorithm 1. Note that this training process partly contains self-learning: $f_\theta$ is trained on the belief set which is in turn partly generated by $f_\theta$ via Eq.(7). This is intended as it is shown to lead to a smooth learning process (Hu et al., 2016).

## 5 EXPERIMENTS

In the experiments, we demonstrate the following properties of LOGICDP: **(P1)** the budget-aware framework empowers LOGICDP to generate high-quality triples leading to better data efficiency than generic weakly-supervised methods; **(P2)** LOGICDP is robust against the incorrect labels; and **(P3)** the logic rule is interpretable, allowing humans to supervise the model efficiently.

### 5.1 DATASETS

**Knowledge graphs**. We evaluate LOGICDP on two KG datasets. FB15K-237 is a subset of the Free-base dataset (Toutanova & Chen, 2015) which contains general knowledge facts. WN18RR (Dettmers et al., 2018) is the subset of WordNet18 which contains relations between English words. We evaluate the LOGICDP on standard graph reasoning on binary predicates with train/valid/test splits provided in (Yang et al., 2017). In addition to it, we remove a large portion of triples from the train split, effectively reducing the size of the bootstrap set $\mathcal{T}_{init}$, so that we can inspect how the size of $\mathcal{T}_{init}$ can influence the performance. And the removed triples are used as the unlabeled sets for baseline methods such as active learning, where the ground-truth labels are hidden. (Statistics in Table 4).

**Visual Genome**. We also evaluate LOGICDP on the Visual Genome (VG) dataset provided in (Krishna et al., 2016) which contains a scene graph for each image. The original dataset is highly noisy (Zellers et al., 2018). We instead use a processed version provided in (Hudson & Manning, 2019). We further process the relational data by filtering out the relation types that occur less than 1500 times. The remaining set consists of 31 types of relations. For each scene graph, we filter out the isolated nodes that are not connected by those relations. For this benchmark, the model is tasked to predict the class label of an entity in the image. To do so, we rank the classes by their frequencies: triples belonging to the top 50 classes together with all the binary predicate triples are treated as background knowledge and given to the model; triples of the next 70 classes are split into bootstrap/valid/test/unlabeled sets (triples of the rest of the classes are discarded as they are too sparse for evaluation). The bootstrap set $\mathcal{T}_{init}$ for each of the 70 classes contains 15 random triples, and the valid, test, and unlabeled sets are split with a 10%/80%/10% ratio on the remaining triples.

Table 1: Training the TransE and the RotatE models with 5%, 10% and 20% of the bootstrap set $\mathcal{T}_{init}$ on two knowledge bases. LOGICDP achieves the best scores **(P1)** and has similar performance as the case trained with full ground-truth data, i.e. the *ideal\** case.

| Methods | TransE | | | | | | | | | | | | RotatE | | | | | | | | | | | |
|---|---|---|---|---|---|---|---|---|---|---|---|---|---|---|---|---|---|---|---|---|---|---|---|---|
| | FB15K-237 | | | | | | WN18RR | | | | | | FB15K-237 | | | | | | WN18RR | | | | | |
| | MRR | | | Hits@10 | | | MRR | | | Hits@10 | | | MRR | | | Hits@10 | | | MRR | | | Hits@10 | | |
| | 5% | 10% | 20% | 5% | 10% | 20% | 5% | 10% | 20% | 5% | 10% | 20% | 5% | 10% | 20% | 5% | 10% | 20% | 5% | 10% | 20% | 5% | 10% | 20% |
| AL-RND | 0.21 | 0.21 | 0.22 | 36.1 | 36.7 | 36 | 0.28 | 0.29 | 0.31 | 28.4 | 32.8 | 36.5 | 0.24 | 0.25 | 0.27 | 37.1 | 39.5 | 41.2 | 0.27 | 0.27 | 0.32 | 29.7 | 34.0 | 43.5 |
| AL-MaxEnt | 0.22 | 0.22 | 0.23 | 36.1 | 37.2 | 36.5 | 0.29 | 0.31 | 0.36 | 29.5 | 35.1 | 45.9 | 0.24 | 0.28 | 0.29 | 38.5 | 43.2 | 47.5 | 0.31 | 0.32 | 0.37 | 33.2 | 42.2 | 51.2 |
| LabelProp | 0.22 | 0.23 | 0.24 | 35.9 | 38.6 | 40.2 | 0.30 | 0.32 | 0.38 | 31.2 | 38.3 | 43.6 | 0.25 | 0.28 | 0.29 | 38.8 | 43.8 | 46.9 | 0.31 | 0.33 | 0.37 | 34.8 | 43.9 | 50.8 |
| LOGICDP | **0.27** | **0.28** | **0.29** | **43.1** | **43.8** | **44.5** | **0.38** | **0.42** | **0.43** | **46.7** | **49.8** | **50.5** | **0.31** | **0.32** | **0.32** | **49.5** | **51.7** | **52.0** | **0.41** | **0.43** | **0.44** | **49.4** | **52.9** | **53.7** |
| *ideal\** | | 0.29 | | | 46.5 | | | 0.45 | | | 51.2 | | | 0.34 | | | 52.6 | | | 0.46 | | | 55.2 | |

## 5.2 Experimental Settings

**Oracle**. We evaluate LOGICDP with two oracle settings: (1) **synthetic**. Experiments in section 5.3 **(P1)** is conducted with a synthetic oracle. We simulate human weak supervision by computing $supp(R)$ on a separate held-out set and sending it back as $\lambda$. (2) **Human**. To better demonstrate the interpretability of LOGICDP **(P3)**, we also evaluate LOGICDP on a fixed set of questions with human oracles who are asked to judge the quality of the rule based on commonsense.

**Graph reasoning models**. For FB15K and WN18RR, we use **TransE** (Bordes et al., 2013) and **RotatE** (Sun et al., 2019) as the reasoning models as they are widely used methods for KG reasoning. For VG, the generated triples are effectively class labels of the scene entities; the task reduces to a multi-class classification task, therefore, we use a standard **MLP** model which takes in the RCNN feature (provided in (Hudson & Manning, 2019)) of the entity and outputs its class label.

**ILP model**. LOGICDP is agnostic to the choice of ILP model. We choose the ILP model based on the type of benchmark dataset. For FB15K and WN18RR, the predicate and entity spaces are moderate, we use the forward-chaining method dNL-ILP (Payani & Fekri, 2019) for rule learning. For VG, the predicate and entity spaces are significantly larger, we use the backward-chaining method NLIL (Yang & Song, 2020) as it is more scalable than the forward-chaining ones. In general, LOGICDP is insensitive to the choice of ILP methods (Appendix F) and performs similarly across different configurations.

**Baselines**. LOGICDP utilizes a budget-aware framework for supervision where we train the model with a bootstrap set $\mathcal{T}_{init}$ and a limited budget $B$. This setting is comparable to those generic weakly-supervised methods. We demonstrate **(P1)** by comparing against 4 generic weakly-supervised methods: (1) **AL-RND**: a naive method that trains $f_\theta$ with ground-truth triples; (2) **AL-MaxEnt**: the active learning method with maximum entropy criterion with budget size as $B$; (3) **LabelProp**: a semi-supervised method that infers labels using label propagation algorithm (Zhu & Ghahramani, 2002); (4) **MAML** (Finn et al., 2017): a meta-learning framework that learns the optimal parameter initialization (this is only evaluated on VG as there is no balanced set of sub-graphs in72 FB15K and WN18RR). Apart from $\mathcal{T}_{init}$, methods (1)(3)(4) are given $B$ randomly sampled ground-truth triples, and method (2) can acquire $B$ ground-truth triples from the unlabeled sets.

**Parameters and evaluation**. We set hyper-parameters $\gamma = 0.7$, and $C = 20$ by running a grid search over the validation set. For FB15K and WN18RR, we evaluate the mean reciprocal rank (MRR) and hits@10. For the scene graph dataset, We evaluate the Recall@1 (R@1) score. Experiments are conducted on a machine with i7-8700K and one GTX1080ti. Implementation available here [1].

## 5.3 Data Efficiency

**Knowledge graphs**. We evaluate the methods by changing the size of the $\mathcal{T}_{init}$ while fixing $B = 5$. We show the results with 5%, 10% and 20% of the original $\mathcal{T}_{init}$ in Table 1. We also show the performance when the model is trained with full ground-truth data, i.e. the *ideal\** case. LOGICDP outperforms all baselines in both two benchmarks. Note that the gap is large for WN18RR because many triples can be inferred with similar logic rules. LOGICDP also achieves similar performance as that in the *ideal\** case with 10% of the data.

**Visual Genome**. We evaluate the methods by varying the budget $B$ from 1 to 500 while fixing $|\mathcal{T}_{init}| = 15$. We show the results in Table 2 and example logic rules in Figure 5. Similar to Table 1,

---

[1] https://github.com/gblackout/logic-data-programming

Table 2: Training the MLP with varied budget sizes on VG dataset. LOGICDP achieves the best R@1 **(P1)** and 80% performance as the case trained with full ground-truth data (*ideal\** case) with $B = 10$.

| R@1 of MLP on Visual Genome | | | | | | |
|---|---|---|---|---|---|---|
| Method w/ MLP | Budget | | | | | |
| | 1 | 5 | 10 | 50 | 200 | 500 |
| AL-RND | 0.47 | 0.47 | 0.5 | 0.52 | 0.57 | 0.6 |
| AL-MaxEnt | 0.47 | 0.47 | 0.51 | 0.52 | 0.58 | 0.61 |
| LabelProp | 0.49 | 0.51 | 0.51 | 0.57 | 0.59 | 0.59 |
| MAML | 0.48 | 0.48 | 0.5 | 0.54 | 0.58 | 0.6 |
| LOGICDP | **0.56** | **0.57** | **0.58** | **0.6** | **0.62** | **0.63** |
| *ideal\** | | | | | | 0.71 |

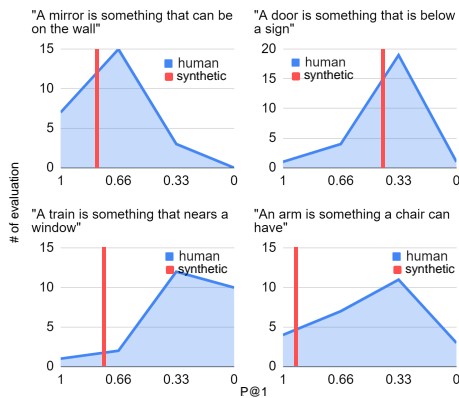

Figure 2: Human evaluation score distributions of 4 questions. The red bar corresponds the score of the synthetic oracle.

LOGICDP achieves the best R@1. The improvement moving from budget 1 to 10 is small because most triples can be generated with less than 3 rules and additional rules typically cover rare triples that only occur less than 50 times for each class. As a verification, we see that LOGICDP achieves 80% of the *ideal\** performance with $B = 10$, meaning most of the correct samples are covered in the belief set. Note that ILP generates at most $|\mathcal{T}_{init}|$ number of rules (i.e. one unique rule per sample), thus for $B > 15$, LOGICDP simply fills in the belief set with random triples just as that in **AL-RND**, in which case, the performance improves at a rate similar to other baselines.

Technically, the budget size of LOGICDP and other methods are not comparable as a logic rule carries more information than a single triple. However, the main motivation of LOGICDP is to train the model efficiently with questions that generalize to many triples. In this way, being able to carry more information in a single query is by itself an advantage over the traditional paradigm. In section 5.5, we further quantify this difference by comparing the efforts spent on evaluating a rule vs. a triple.

### 5.4 ANALYSIS ON LABELING ERRORS

Imperfect logic rules introduce noisy labels into the belief set. We analyze the robustness of LOGICDP against these errors **(P2)** on VG dataset with $B = 5$. In Figure 3, we show the ratio of false positives (FP), false negatives (FN), and true positives (TP) for each of the 70 classes with respect to the size of their ground-truth triples.

Ideally, if one learns a perfect set of rules for a class $P$, then the TP ratio should be $100\%$ and FP and FN be zero. However, a noisy rule can introduce (1) FP, where triples of other classes are incorrectly classified as $P$. Introducing FP deteriorates the belief set and is harmful to the training. Here, figure 3 suggests that the FP ratio is small across all classes and the total FP ratio is 6.7%, suggesting the rules are of high support. On the other hand, there are (2) FN, where ground-truth triples of class $P$ are not covered by any rules and thus fall outside of the belief set. We found that logic rules are generally of high coverage for frequent classes and less for infrequent ones. This suggests that these classes contain more diverse graph patterns and the rules learned from a small bootstrap set cannot cover all of them. This aligns with the fact that VG contains massive scene graphs of various structures. Unlike FP, FN does not deteriorate the performance and one can readily improve the coverage by increasing the size of the bootstrap set.

### 5.5 WEAK SUPERVISION VIA INTERPRETABLE LOGIC RULES

As one of the key motivations, we argue that human supervision via logic rules is data-efficient because humans can evaluate rules directly with commonsense **(P3)**. To see this, we compare two types of supervision paradigms: (1) **sample-wise labeling**, where humans supervise the model by labeling individual samples; and (2) **rule-wise labeling**, where humans supervise the model by evaluating the rules (i.e., the function refinement introduced in section 4.2).

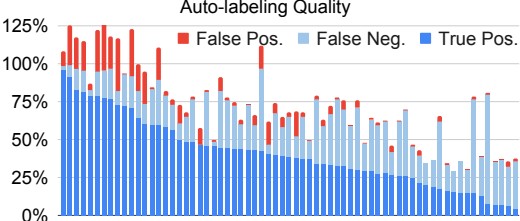

Figure 3: The class-wise ratio of true positive, false negative, and false positive triples.

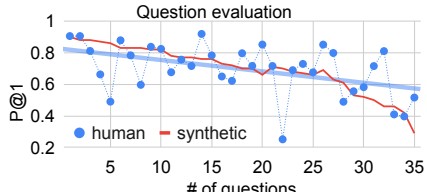

Figure 4: P@1 scores of the 35 logic questions sorted in descending order with respect to the human score.

For this study, we invite 25 graduate students to go through these two paradigms on VG dataset and record the "wall clock time taken" for completing the task (Details in Appendix E). For sample-wise labeling, the participant is shown 50 images randomly sampled from the target classes. Target objects are highlighted with the bounding boxes and the participant is asked to mark true or false on whether or not the image patch belongs to the target class. For rule-wise labeling, We collect 35 logic rules and convert them into natural language questions (Figure 6). For each question, the participants choose from a 4-point scale that reflects how they think the rule is generally true or not; we interpret this as the intuitive support score $supp(R)$.

Table 3: Average/amortized time taken in evaluating the individual samples and the logic rules.

| Eval. type | Avg. time (s) | Std. (s) |
|---|---|---|
| Sample | 3.7 | 1.8 |
| Rule | 8.3 | 3.1 |
| Amortized | **0.03** | - |

We summarize the average labeling time of two types of supervision in Table 3. Rule-wise labeling takes 8.3s which is 2x longer than the sample-wise labeling, but this time is amortized as the rule applies to many samples, leading to an amortized time of 0.03s, which is 100x faster than the sample-wise counterpart.

**Synthetic oracle vs. human oracle**. We compare the scores of two oracles of the 35 questions in Figure 4. The 4-point scores of the human oracle are converted to $\{1, 0.66, 0.33, 0\}$ respectively and are averaged across participants for each question. Overall, the human evaluations tend to agree with that of the oracle: we fit the scores with a linear trend line (in light blue) and it is close to the score curve of the oracle. However, there are also outliers: Figure 2 showcases the score distributions of four questions, where the two oracles agree on the first row but disagree on the second row. This is due to the inherent bias in the dataset: in real-life, a Chair does not always have arms, but in VG, the Arm is highly correlated with Chair in the dataset, leading to counter-intuitive statements. Addressing this issue involves altering the original dataset and we leave this for future investigation.

Additionally, Figure 2 also suggests that human evaluations are generally consistent for different participants. This observation is verified in Appendix E, where we found the standard deviations of the scores are small. This aligns with **(P3)** that human commonsense is a reliable source of weak supervision for tasks such as image classification. Nevertheless, for domains that involve graphs of complex reasoning or procedures, such as theorem proving and molecular synthesis, human commonsense will not be as effective. In future work, we investigate extending the framework to these domains.

## 6 CONCLUSION

In this work, we propose LOGICDP, a data programming framework optimized for creating labels for graph data. The framework is capable of automatically learning labeling functions in the form of logic rules via ILP and refining them with a budget-aware interactive framework. Experiments show that LOGICDP is more data-efficient than generic weakly-supervised methods and supports efficient human-in-the-loop supervision.

## ACKNOWLEDGMENTS

This work was supported in part by a sponsored research award by Cisco Research.

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

## A  LOGIC RULES AND INDUCTIVE LOGIC PROGRAMMING

A first-order logic (FOL) rules consists of (i) a set of predicates defined in $\mathcal{P}$, (ii) a set of logical variables such as $X$ and $Y$, and (iii) logical operations $\{\wedge, \vee, \neg\}$. For example,

$$\texttt{GrandFatherOf}(X, X') \leftarrow \texttt{FatherOf}(X, Y) \wedge \tag{8}$$
$$\texttt{MotherOf}(Y, X')$$

involves predicates `GrandFatherOf`, `FatherOf` and `MotherOf`. Components such as `FatherOf`$(X, Y)$ are called **atom**s which correspond to the predicates that apply to the logical variables. Each atom can be seen as a lambda function with its logical variables as input. This function can be evaluated by **instantiating** the logical variables such as $X$ into the object in $\mathcal{X}$. For example, let $\mathcal{X} = \{\text{Amy}, \text{Bob}, \text{Charlie}\}$, we can evaluate `FatherOf`$(\text{Bob}/X, \text{Amy}/Y)$ by instantiating $X$ and $Y$ into Bob and Amy respectively. This yields 1 (i.e. `True`) if "Bob is the father of Amy". The outputs of all atoms are combined using logical operations $\{\wedge, \vee, \neg\}$ and the *imply* operation $p \leftarrow q$ is equivalent to $p \vee \neg q$. Thus, when all variables are instantiated, the rule will produce an output as the specified combinations of those from the atoms. By using the logical variables, the rule encodes the "lifted" knowledge that does not depend on the specific data. Such representation is beneficial because (i) the rules are highly interpretable and can be translated into natural language for human assessment, and (ii) the rules guarantee to generalize to many examples.

In section 4.1, we learn a set of rules $\mathcal{R}_{cand}$ from bootstrap set $\mathcal{T}_{init}$ via ILP. Given a query triple $\langle x, ?, x' \rangle$, ILP seeks to learn a logic rule that captures a graph pattern within which the query triple is `True`. This can be formulated as finding a relational path $x \xrightarrow{P^{(1)}} \dots \xrightarrow{P^{(T)}} x'$ which starts from one node $x$ and ends at the other node $x'$, where $P^{(t)} \in \mathcal{P}, t = 1, \dots, T$. Finding such path is equivalent to learning a chain-like entailment rules (Yang et al., 2017) as

$$R : P(x, x') \leftarrow P^{(1)}(x, z_1) \wedge P^{(2)}(z_1, z_2) \dots \wedge P^{(T)}(z_{T-1}, x'). \tag{9}$$

where the knowledge is encoded as "if the path exists, then $P(x, x')$ is `True`". Numerically, we have $R : \mathcal{X} \times \mathcal{X} \mapsto \{0, 1\}$, that is $R(x, x') = 1$ if the query is `True`. Note that if the path does not exist, the rule outputs $R(x, x') = 0$ or *abstain*. Due to the nature of logical implication "$\leftarrow$", "abstain" simply means the rule does not apply instead of logical `False`. In other words, the rule $R$ can be used to infer positive triples but not negative ones. Since many of the KG completion models are optimized with margin loss Eq.(1) and thus do not rely on the negative triples for training, we omit the learning of the negative rules. This chain-like rule is also applicable to unary predicate $P(X)$ by instantiating $X'$ into a localized entity (see details in (Yang & Song, 2020)).

To compute the path efficiently, one encodes a graph into a set of adjacency matrices. Each predicate $P$ can be represented as $\boldsymbol{M} \in \{0, 1\}^{|\mathcal{X}| \times |\mathcal{X}|}$, where $m_{ij} = 1$ indicates $\langle x_i, P, x_j \rangle$ exists in the graph. In the case of unary predicate, the matrix is diagonal such that $m_{ii} = 1$ indicates $\langle x_i, P, x_i \rangle$ exists in the KG. Here, we duplicate the same variable for notation consistency. Recall that $v_x$ is the one-hot vector with dimensionality $|\mathcal{X}|$. We can represent each hop in the relational path as matrix multiplication, such that the $(t)$th hop of the reasoning along the path is computed as

$$v^{(0)} = v_x, \qquad\qquad v^{(t)} = v^{(t-1)\top} \boldsymbol{M}^{(t)},$$

where $\boldsymbol{M}^{(t)}$ denotes the adjacency matrix of the predicate used in $(t)$th hop. One can verify that the $j$th element $v_j^{(t)}$ in $v^{(t)}$ is the count of unique paths from $x$ to $j$ (Guu et al., 2015). After $(T)$ steps of reasoning, we compute the score of $P(x, x')$ being `True` as

$$\text{score}(v_x, v_x') = v_x^\top \prod_{t=1}^{T} \boldsymbol{M}^{(t)} \cdot v_x'. \tag{10}$$

Finding the chain-like rule is equivalent to finding a sequence of matrix multiplication $\boldsymbol{M}^{(t)}, t = 1, \dots, T$, such that Eq.(10) is maximized for all the triples in $\mathcal{T}_{init}$. This hard search problem can be relaxed into learning the weighted sums of all possible paths

$$\text{score}(v_x, v_x' | \alpha, \beta) = v_x^\top \sum_{t'=1}^{T} \alpha^{(t')} \left( \prod_{t=1}^{t'} \sum_{k=1}^{K} \beta_k^{(t)} \boldsymbol{M}_k \cdot v_x' \right), \tag{11}$$

where $\alpha = [\alpha^{(1)}, ..., \alpha^{(T)}]^\top$ is the path attention vector, and $\beta^{(t)} = [\beta_1^{(t)}, .., \beta_K^{(t)}]^\top$ is the matrix attention vector at $(t)$th step, and we denote all the attentions as $\langle \alpha, \beta \rangle$. Therefore, for each triple $P(x, x')$ one can find the optimal logic rule by maximizing Eq.(11) with respect to $\langle \alpha, \beta \rangle$. To extract the actual discrete rule, one takes the $\texttt{ArgMax}$ on $\langle \alpha, \beta \rangle$, which returns the index of the max value. The optimal rule length is $T^* = \texttt{ArgMax}(\alpha)$ and the $(t)$th matrix is $\boldsymbol{M}^{(t)} = \texttt{ArgMax}(\beta^{(t)})$ which corresponds to the predicate $P^{(t)}$.

These attentions can be generated differentiably by deep models. For example, NeuralLP (Yang et al., 2017) uses an RNN controller to generate the sequence of attention vectors with $x$ as the initial input. And NLIL (Yang & Song, 2020) generates attentions with a stacked Transformer module with predicate embeddings as the input. Let $\texttt{ILP}_\omega$ be the differentiable ILP model, we optimize paramter $\omega$ such that

$$\arg\max_\omega \sum_{\langle v_x, P, v'_x \rangle \in \mathcal{T}_{init}} \text{score}(v_x, v'_x | \alpha, \beta), \tag{12}$$

where $\langle \alpha, \beta \rangle = \texttt{ILP}_\omega(\langle v_x, P, v'_x \rangle | \mathcal{G})$. The LOGICDP framework is agnostic to the actual choice of the ILP model, one can also apply forward-chaining methods such as $\partial$ILP (Evans & Grefenstette, 2018) or dNL-ILP (Payani & Fekri, 2019) to achieve similar outcomes. Finally, we collect all rules learned from $\mathcal{T}_{init}$ as our rule set $\mathcal{R}_{cand}$.

## B  POSTERIOR REGULARIZATION

In section 4.3, we apply the *posterior regularization* (Ganchev et al., 2010) on the rule set and construct $\mathbf{M}_{\text{label}}$ as a *teacher* model that aggregates the joint inferences of the rules in $\mathcal{R}$. Formally, a teacher model is defined as the solution to a convex optimization problem

$$\min_{\mathbf{M}_{\text{label}}, \boldsymbol{\xi}} \text{KL} \left( \mathbf{M}_{\text{label}}(\boldsymbol{t}|\mathcal{R}) \| f_\theta(\boldsymbol{t}) \right) + C \sum_{j=1}^{B} \xi_j, \tag{13}$$

$$s.t. \; \lambda_j (1 - \mathbb{E}_{\mathbf{M}_{\text{label}}} [R_j(x, x')]) \leq \xi_j, \; j = 1, ..., B, \tag{14}$$

where $\boldsymbol{\xi} = \xi_1, ..., \xi_B \geq 0$ are the slack variables and $C$ is the regularization coefficient controlling the weight of the penalty term. Intuitively, we want the teacher to reflect the belief of the logic rules. This is done by putting the inequality constraint Eq.(14). On the other hand, while rule constraints are all satisfied, we want the teacher to stay close to $f_\theta$ which is the reasoning model we seek to train, such that it creates a smooth optimization surface that makes it easy for the $f_\theta$ to learn from the teacher. This is done by minimizing the KL divergence in Eq.(13) between $\mathbf{M}_{\text{label}}$ and $f_\theta$. This problem is solved analytically

$$\mathbf{M}_{\text{label}}(\boldsymbol{t}|\mathcal{R}) \propto f_\theta(\boldsymbol{t}) \exp \left( -C \sum_{j=1}^{B} \lambda_j (1 - R_j(x, x')) \right), \; \boldsymbol{t} := \langle x, P, x' \rangle, \tag{15}$$

## C  DATASET STATISTICS

We summarize the statistics of the three datasets in Table 4.

Table 4: Statistics of FB15K-237, WN18RR and Visual Genome datasets.

| KG | # facts | # entities | # predicates |
|---|---|---|---|
| FB15K | 272K | 15K | 237 |
| WN18RR | 93K | 41K | 11 |
| VG | 1.9M | 1.4M | 2100 |

## D  EXAMPLE LOGIC RULES

We show example logic rules learned from the Visual Genome dataset in Figure 5.

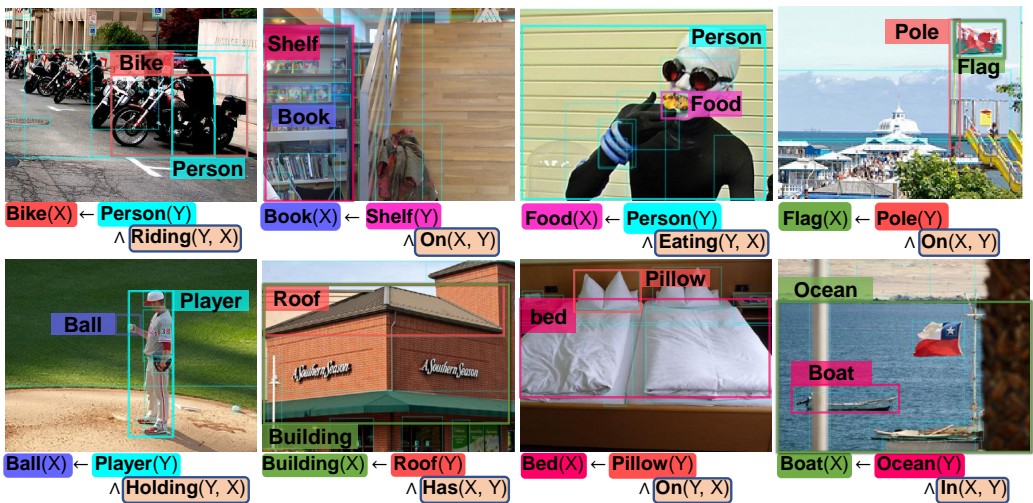

Figure 5: Showcases of logic rules learned from the scene graphs.

# E  LOGICDP WITH HUMAN ORACLE

A mirror is something that can be on the wall *
○ Yes, always
○ Yes, usually
○ Not quite right, but sometimes yes
○ No

A mirror is something that can be on the motorcycle *
○ Yes, always
○ Yes, usually
○ Not quite right, but sometimes yes
○ No

A glass is something that can be on the table *
○ Yes, always
○ Yes, usually
○ Not quite right, but sometimes yes

Figure 6: Screenshot of the question forms

The screenshot of the question forms used in the human oracle study is shown in Figure 6. This study is approved by the institute IRB. The document will be made available after the review phase for anonymous purposes.

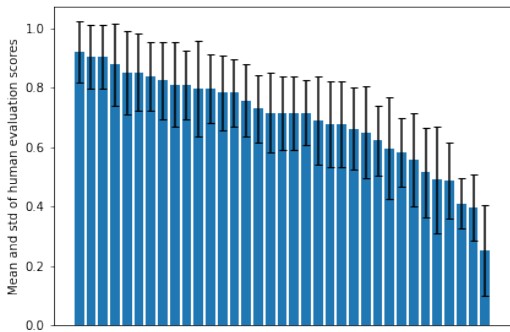

Figure 7: Mean and std of human evaluation scores of the 35 questions collected in section 5.5.

**Human score distribution**. We show the mean and std of human evaluation scores in Figure 7 for the 35 questions. We find the confidence scores are generally consistent for all participants, which suggests that the logic rules can be evaluated with general human commonsense. However, note that LOGICDP does not require a consistent score distribution to function: each rule $R$ obtains only a single confidence score $\lambda$.

## F    ADDITIONAL ABLATION STUDIES

Table 5: Performance of LOGICDP with three different ILP methods on the FB15K and VG datasets: NeuralLP (Yang et al., 2017), NLIL (Yang & Song, 2020), and dNL-ILP (Payani & Fekri, 2019).

| ILP method | FB15K-237 | | VG |
| | MRR | Hits@10 | R@1 |
| --- | --- | --- | --- |
| NeuralLP | 0.26 | 38.9 | 0.55 |
| NLIL | 0.28 | 42.9 | 0.57 |
| dNL-ILP | 0.28 | 43.8 | 0.54 |

LOGICDP is agnostic to the choice of ILP methods. In the experiments, we found LOGICDP performs similarly with both dNL-ILP (Payani & Fekri, 2019) and NLIL (Yang & Song, 2020) methods, indicating LogicDP is also insensitive to the choice of ILP methods. To further validate this observation, we show LOGICDP performance on FB15K and VG datasets with three ILP methods. We use the same budget and bootstrap set size as in section 5.3. For FB15K, we set $|\mathcal{T}_{init}|$ to $10\%$ and $B = 5$; for VG, $|\mathcal{T}_{init}| = 15$ and $B = 5$. The results are shown in Table 5, which suggests that LOGICDP is not sensitive to the choice of the ILP methods.

