# OpenReview forum: "LogicDP: Creating Labels for Graph Data via Inductive Logic Programming"
_ICLR.cc/2023/Conference — ICLR 2023 poster_

### Official Review · Reviewer_FEgy · 2022-10-23

**Confidence:** 3
**Correctness:** 3
**Technical Novelty And Significance:** 3
**Empirical Novelty And Significance:** 3
**Recommendation:** 6

**Clarity, Quality, Novelty And Reproducibility:**

The model is well-positioned in relation to existing works. Using ILP for DP is (as far as I know) a novel and exciting idea. The technical execution and evaluation seem solid, modulo my clarity challenges.

Logic rules as labeling functions: "...which encodes the statement that P(x,x') is True if and only if the body ... is True" - I don't think this is correct, the Horn-like rule has a simple implication (not a bidirectional iff)?

Baselines: "LogicDP utilizes a budge-aware framework" - typo/misspelling of "budget".

Figure 3: it would be helpful to have an in-place x-axis label here.

4.2 Function refinement:
-probably it is variation in terminology across sub-fields, but I mentally mapped supp() to be the "precision" of the rule R.
-also, are the predicates typed here? If not, |X|^2 is always the same for all R and is ignored in practice for cov(), right?

Algorithm 1: for the belief set update based on Eqn 5, are we adding _exactly_ one triple <x,P,x'> each outer loop? The notation
here got a little unclear for me. If so, why not add the Top N triples under M(t|R)?

5.4 Analysis on labeling errors:
-"6.7% of the ground-truth triples are FP" - this doesn't make any sense to me?
-"indicates that LogicDP learns high-quality functions" - is it accurate to characterize the learned rules as "high precision, low/medium recall"?
-in Figure 3, what explains the persistently high proportion of False Negative? Are there "difficult" triples to cover with rules? If so, are there any common or interesting properties of these?

5.5 Weak supervision via interpretable logic rules:
-"...humans can evaluate rules directly with commonsense." - I would suggest to qualify this statement, arguably it is true for some domains (eg, wheels on cars) but probably not others (eg, binding sites on proteins).
-The paper is not misleading here, but I think it would be fair to emphasize that of course the human experimenters did the work of converting the logic rules into natural language statements for the human labeling oracles - automating this could be called out as some future work.

Figure 3 and 4: I had to flip back and forth between the text and these images a lot to figure out exactly what was going on. I'm not sure if there is an easier or clearer way to visualize the relationships here, but as-is I think these figures put some heavy lifting on the reader.

Appendix D: I guess Figure 5 is the content, but the section could have some text indicating this.

**Strength And Weaknesses:**

The core idea is a compelling, natural, and creative extension of existing research directions. The technical execution of the idea seems sound, and the experimental evaluations show some good advantages of the approach.

I found some clarity challenges in both the method description and the evaluation. To some extent this is probably an inherent challenge to the "model within a model" approach and the varied techniques (logic rules, neural nets, continuous relaxations, posterior regularizations) employed here. It might be helpful to have a running concrete example, or more diagrams with a concrete example for different steps in the process. Besides this I had a number of smaller stumbling blocks from a clarity perspective, documented below.

Baselines: why not compare against a "Data Programming" method like Snorkel or Snuba? It seems strange that only AL-MaxEnt is also doing a budget-based approach (although I guess it makes sense to give the other methods B extra randomly labeled triples).

Given the "self-training" in the approach, are there potentially catastrophic failure modes if poor initial rules or triples are chosen?

The "outliers" in the synthetic-vs-human oracle due to bias in the dataset seems like a valuable observation and good direction for future work.

**Summary Of The Paper:**

This paper proposes an integrated prediction and labeling framework for graph data based on learning logical labeling rules which are scored by an oracle. Candidate labeling rules are learned via Inductive Logic Programming (ILP), and new labeled triples are identified using the potentially noisy rules via posterior regularization. Experimental evaluation show LogicDP achieves good performance with fewer original labeled data points by using the labeling rules, and that, for human oracles, labeling rules for these tasks is much more time-efficient than labeling individual instances.

**Summary Of The Review:**

I find the core contribution to be strong, although there is some room for improvement on clarity aspects.

---

> ### Author Response · Authors · 2022-11-12
> **Response to FEgy**
>
> We thank the reviewer for the comments. Our responses are as follows:
>
>
> **Comparing against existing data programming methods**
>
> We thank the reviewer for the suggestion. We have considered comparing LogicDP to Snorkel and Snuba but found it very difficult to set up a fair benchmark. Snorkel requires hand-written labeling functions. They are expensive to collect and suppose they are collected, it is difficult to quantitatively compare the final performance against the amount of effort spent. For example, it is unclear what would be the budget for writing a function from scratch vs evaluating an existing function.
>
> Comparing to Snuba appears to be a better option because it can automatically learn labeling functions. However, the current implementation only supports learning decision tree-like functions from flat feature vectors. The framework requires some heavy modifications to fit into the graph reasoning pipeline. For example, replacing flat features into subgraphs and designing a labeling function family to be learned from graphs, in which case it effectively becomes a reinvention of LogicDP.
>
> Therefore, we eventually decided to compare LogicDP with generic weakly-supervised methods. The rationale is three-fold: (1) the pipeline of LogicDP resembles the standard active learning paradigm, where an initial training set is given and the model actively asks for information under a limited budget; (2) one can quantify the "effort spent" as the number of budgets for a sensible comparison. Although, as discussed in the last paragraph of 5.3, $B$ rules certainly carry more information than $B$ triples, but (3) we believe this in fact demonstrates the advantage of LogicDP and data programming methods in general. Section 5.5 illustrates this by comparing the wall clock time spent with these two labeling paradigms in a more realistic and fair setting, and we found that the amortized labeling time of LogicDP is 100x less than direct labeling.
>
>
> **Budget for the baselines**
>
> We apologize for the confusion. All baseline methods are indeed doing the same budget-based approach. [Baselines paragraph of section 5.2]: all baseline methods can access up to $B$ samples for each predicate. In particular, AL-MaxEnt is given $B$ queries and the access to the unlabeled set, and the rest of the methods are given $B$ random ground-truth samples as they don't require actively collecting samples.
>
>
> **Can self-training lead to catastrophic failure modes?**
>
> Thank you for the great insight. Yes, the framework can partially fail when a bad bootstrap set with incorrect labels is given, but it can be easily detected and then fixed. A bad bootstrap set effectively leads to a bad rule candidate set with many incorrect rules. However, since rules are scored by the oracle, if we are willing to assume human oracles have good heuristics (as do any general DP methods), a low score will be given and one can simply filter them out by setting a score threshold during the refinement phase. If all rules in the candidate set $\mathcal{R}_{cand}$ are filtered out, one can fix it either by correcting the bootstrap set or manually modifying the rules.
>
>
> **Support and coverage definition**
>
> Yes, support and coverage are terms typically used in knowledge base rule mining. And they are computed separately for each different predicate type. In our setting, support is similar to precision, whereas coverage measures how many samples this rule could cover. It is different from recall because we do not know the ground-truth set triples of predicate type $P$.
>
> **Belief set updates in Eq 5**
>
> At each iteration, we first update the labeling model $M_\text{label}$, then infer the predicate labels for every possible pair of entities; all the triples generated here are collected and combined with the bootstrap set, which yields the belief set at this iteration (Algorithm 1). Therefore, there are many, instead of one, triples generated in Eq 5. Theoretically, the number should be $|\mathcal{X}|^2$, but in practice, this is prohibitive for large KGs such as FB15K, and we follow the convention and infer labels for a fixed set of triples whose ground-truth predicates are hidden, that is the unlabeled set.
>
> We have revised the section to clarify this aspect.

---

> ### Author Response · Authors · 2022-11-12
> **Response to FEgy cont'd**
>
> **Figure 3 and the analysis on labeling errors**
>
> Figure 3 shows the TP, FP, and FN rates for each of the 70 classes. Since the x-axis here is the class ID, we sort them by TP rate for better clarity.
>
> One can understand Fig 3 in the following way: given a class $P$ whose set of ground-truth triples is $\mathcal{T}^*_P$:
> - TP is the ratio of triples $\in \mathcal{T}^*_P$ that are correctly labeled as $P$ by rules. Thus it is between 0~100%.
> - FP is the number of triples $\notin \mathcal{T}^*_P$ but are incorrectly labeled as P divided by $|\mathcal{T}^*_P|$. Thus, TP+FP can be over 100%.
> - FN is the ratio of triples $\in \mathcal{T}^*_P$ that are not labeled as $P$ by rules. This corresponds to the case where none of the rules apply to the triple, and thus, the predicate cannot be inferred.
>
> Therefore, "6.7% of the ground-truth triples are FP" means there are 6.7% of the triples are incorrectly labeled and the high FN rate for the longtail classes suggests that there are many triples that cannot be covered by the rules. This happens when the local graph patterns in these classes are diverse and, therefore, the set of rules proposed from a small bootstrap set cannot cover all of them. This is pretty common in VG dataset because it contains massive noisy crowdsourced scene graphs, and the patterns can change significantly for some longtail classes.
>
> And yes, Fig 3 also aligns with our design in 4.2, where we favor rules of high precision over high recall, because FP is more harmful to learning. To reach better recall, one can simply increase the size of the bootstrap set, or design a schema that also updates the bootstrap set during refinement to keep proposing new rules. However, we presume this process leads to a more complex self-training scenario, which requires some careful designs. We leave this for future work.
>
> We have updated 5.4 to clarify this aspect.
>
> **Typos, errors, and discussions**
>
> We thank the reviewer for the comments. We have made the following changes to the draft:
> - Fixed the introduction to the Horn-like rule, where it encodes implication instead of the iff condition
> - Added discussion on clarifying that human commonsense is limited to certain domains
> - Adjusted the position of Fig 3, 4 and 5

---

### Official Review · Reviewer_1tSH · 2022-10-24

**Confidence:** 4
**Correctness:** 4
**Technical Novelty And Significance:** 2
**Empirical Novelty And Significance:** 3
**Recommendation:** 5

**Clarity, Quality, Novelty And Reproducibility:**

The clarity and presentation quality of this paper could be improved. This work is practical and reproducible but has limited novelty.

**Strength And Weaknesses:**

### Strength:
- The proposed approach uses ILP to learn human-understandable rules to perform the edge completion task, making human-aided model revision possible.
- The first-order representation reduces human effort in label revision in data programming.
- The proposed approach is straightforward and practical.

###  Weakness:
- Edge completion in knowledge graph by rule induction is a commonly used technique.
- In order to perform rule learning with ILP, users need to prepare a set of primitive predicates and corresponding knowledge base, which increases the difficulty in deploying your method.
- The presentation of this paper could be improved, the introduction to ILP is not enough and the terminology in this paper is confusing. For example, Eq. 5 defines an immediate consequence operator (which is usually denoted as $\mathrm{T_P}$ in logic programming literature) $\mathcal{T}_i$, however, the $i$ does not appear in the RHS of the equation, or maybe it should be $\mathcal{T}_1$?

**Summary Of The Paper:**

This paper presents a weakly supervised learning approach to generate labels for unlabeled triples on graphs. Overall, the proposed approach could be regarded as a combination of inductive logic programming and active learning. The ILP module learns a set of (noisy) logic rules to perform relation completion in the graph data; while the active learning module combines rule coverage and true positive rate into a scoring function to rank the rules and generates queries to ask the oracles to make annotations/revisions.  The combination of ILP and active learning benefits from first-order logic's interpretability and expressive power, which makes human-involved model revision possible, and the revision of first-order logic rules is much more efficient than example-level label revision.

**Summary Of The Review:**

This paper combines ILP and active learning to solve the edge completion task in graph reasoning. The technical novelty of this paper is limited since it is a combination of two mature systems for solving the task, therefore I think this work will attract more audience in data-mining-related venues.

---

> ### Author Response · Authors · 2022-11-12
> **Response to 1tSH**
>
> We thank the reviewer for the comments. Our responses are as follows:
>
>
> **Edge completion in knowledge graph by rule induction is a commonly used technique.**
>
> Yes, ILP is a well-studied technique for graph completion, and we believe this does not undermine the contribution of this work.
>
> LogicDP is not an ILP-based approach for graph completion. The goal of this work is to investigate ILP for data programming on graph data and to the best of our knowledge, LogicDP is the first of this kind.
>
>
> **Need to prepare a set of primitive predicates and corresponding knowledge base**
>
> The aim of this work is to generate labels for graph data such as scene graphs and KGs, in which case the datasets (e.g., Visual Genome, Freebase, and Wordnet) already come with predefined predicates, and no additional predicate invention is needed for applying LogicDP.
>
> We presume the reviewer suggested a different use case, that is to automatically learn both the graph representation and the corresponding logic rules from unstructured data, such as raw text.
>
> While we agree that this is an exciting and interesting direction, it involves challenges that take more than one paper to solve, for example, natural language to first-order logic translation. Nevertheless, we do not regard this as a limitation: there exists a wide range of realistic and important applications within the KG domain such as graph completion, entity resolution, and reasoning that demand high-quality labels, and LogicDP can be readily applied to them for better data efficiency.
>
>
> **Definition of $\mathcal{T}_i$ in eq (5)**
>
> $\mathcal{T}_i$ denotes the belief set at $i$-th iteration. It is a set of triples with inferred predicate label and not a consequence operator from the logic programming literature.
>
> $i$ does not appear on the right-hand side because it is not a direct update from its previous state (say, $\mathcal{T}_{i-1}$), but the output of the labeling model $M_\text{label}$. This set is different at each iteration, because, as shown in Algorithm 1 and section 4.3, $M_\text{label}$ is updated at each iteration.
>
> We have updated the draft to clarify this aspect.
>
>
> **Presentation of ILP**
>
> We apologize for the condensed introduction to ILP. This work does not aim to innovate on the ILP front, but on how the logic rules can be used for automated data programming. Considering the page limit and that this paper lies at the intersection of multiple domains: data programming, ILP, and weak supervision, we decided to leave a detailed ILP discussion in Appendix A. On the other hand, we'd like to also note that, as stated in 4.1, the rule generation is a self-contained standard ILP problem, and LogicDP approaches this by calling existing ILP methods.

---

### Official Review · Reviewer_nzQK · 2022-10-25

**Confidence:** 4
**Correctness:** 2
**Technical Novelty And Significance:** 2
**Empirical Novelty And Significance:** 2
**Recommendation:** 3

**Clarity, Quality, Novelty And Reproducibility:**

The quality and clarity of the paper is good. The descriptions about the proposed approach and experiments are generally clear, except a lack of principles or guidelines on method choices in pipeline steps. The originality is marginal by considering that the proposed pipeline approach is rather straightforward and it is merely a simple application of the data programming (Ratner et al., 2016) framework. No source code or data is provided through the supplemental material.

**Strength And Weaknesses:**

Strengths:

(1) The paper proposes a new method for adding labels to unlabeled graph data, which will be a good contribution to the field of semi-supervised learning.

(2) The advantages of the proposed method are demonstrated with certain choices of the pipeline steps.

Weaknesses:

(1) The experiments are somewhat unfair. The oracle of the proposed approach assigns confidence scores to B logical rules, while the oracles of all other compared methods handle B ground-truth triples. Considering that logical rules are more informative than triples, the superiority of the proposed methods may possibly be due to more information obtained from oracles.

(2) It is unclear whether human beings can assign reasonable confidence scores to logical rules. The experiments do not confirm this. Although the evaluation cost (in time) is reported, there is no evidence to show that different human beings can assign consistent confidence scores to logical rules.

(3) The ultimate performance of the proposal heavily depends on the pipeline steps, including which method is used to learn logical rules and how accurately the oracle assigns confidence scores to selected rules. No principles or guidelines are provided to design a reasonable pipeline step.


**Summary Of The Paper:**

The paper proposes a pipeline approach to supplement labels for graph data. Firstly, it learns some logical rules from existing labeled data. Secondly, it refines the learnt set of logical rules by asking an oracle (such as human being) to assign confidence scores to a fixed number of selected rules. Finally, it estimates labels (the predicates on entities) through the refined set of soft rules and a self-learnt reasoning model. Experimental results on three benchmark datasets demonstrate the proposed approaches outperform four baseline weakly-supervised methods.

**Summary Of The Review:**

Although the paper proposes a new method for adding labels to graph data and provides evaluation results to demonstrate its effectiveness, the originality is marginal considering that the proposal is only a simple application of the data programming framework. Besides, there remain some issues on whether oracles can assign reasonable confidence scores to logical rules and on whether the empirical comparison with other baseline methods is fair enough.

---

> ### Author Response · Authors · 2022-11-12
> **Response to nzQK**
>
> We thank the reviewer for the comments. Our responses are as follows:
>
>
> **The experiments are somewhat unfair: $B$ rules vs $B$ triples**
>
> This is a great insight into our work. In fact, we believe this "unfairness" is the key strength of our approach (and DP methods in general) compared to the traditional weakly-supervised method.
>
> As stated in the last paragraph of 5.3, a logic rule indeed carries more information than a single triple. The benefit is that the effort needed for evaluating a rule does not grow proportionally to the number of samples it labels -- the more samples it labels, the less per-sample labeling time is achieved.
>
> This advantage, however, could not be solely quantified by the number of budgets used, because it does not take into account the "amount of effort" spent on evaluating the query. This is why, in section 5.5, we study a more realistic setting and compare the wall clock time spent with two labeling paradigms respectively. Table 3 shows that the amortized per-sample labeling time is 100x less than direct labeling, which aligns with our claim.
>
>
> **It is unclear whether human beings can assign reasonable/consistent confidence scores to logical rules. And experiments do not confirm this.**
>
> We'd like to answer this question from three aspects:
>
> **(A1)** Our experiment results suggest that human scores are indeed consistent
>
> We observe fairly consistent score distributions in our human evaluation experiments (section 5.5). In Figure 2, we show score distributions of four classes and one can see that scores are generally aligned.
>
> However, we do agree with the reviewer that this aspect can be made more explicit. Here, we show the mean and std of human evaluation scores.
>
> |      |      |      |      |      |      |      |      |      |      |      |      |      |      |      |      |      |      |      |      |      |      |      |      |      |      |      |      |      |      |      |      |      |      |      |      |
> |------|------|------|------|------|------|------|------|------|------|------|------|------|------|------|------|------|------|------|------|------|------|------|------|------|------|------|------|------|------|------|------|------|------|------|------|
> | Mean | 0.72 | 0.68 | 0.72 | 0.73 | 0.25 | 0.66 | 0.85 | 0.40 | 0.49 | 0.82 | 0.84 | 0.49 | 0.78 | 0.81 | 0.91 | 0.85 | 0.91 | 0.65 | 0.92 | 0.80 | 0.69 | 0.41 | 0.56 | 0.72 | 0.58 | 0.80 | 0.81 | 0.68 | 0.60 | 0.52 | 0.88 | 0.72 | 0.62 | 0.78 | 0.76 |
> | Std  | 0.12 | 0.14 | 0.14 | 0.11 | 0.15 | 0.14 | 0.13 | 0.11 | 0.13 | 0.13 | 0.12 | 0.18 | 0.11 | 0.12 | 0.11 | 0.14 | 0.11 | 0.16 | 0.10 | 0.11 | 0.15 | 0.08 | 0.16 | 0.12 | 0.12 | 0.16 | 0.14 | 0.14 | 0.17 | 0.15 | 0.14 | 0.11 | 0.12 | 0.13 | 0.12 |
>
> We have added the discussion to the draft accordingly.
>
>
> **(A2)** LogicDP does not require different humans to assign consistent scores for logic rules, just as Snorkel does not require different humans to write the same labeling function
>
> LogicDP obtains only a single confidence score for each rule and aggregates the outputs. In section 5.5, we show score distribution in order to compare the synthetic oracle with the human one and they are not required in running LogicDP.
>
>
> **(A3)** The criticism that "human-generated score/function can be wrong" is true for all DP methods
>
> Indeed, there is no theoretical guarantee that the scores would be always reasonable. However, the same criticism also holds true for any DP methods in general. Take Snorkel for example, there is no guarantee that humans can always write high-quality labeling functions. Yet, one can still successfully generate high-quality labels with them.
>
> We agree this would be an interesting direction, but we conjecture that solving this challenge would involve joint efforts from both the ML and the cognitive science community, and it is beyond the scope of this work.

---

> ### Author Response · Authors · 2022-11-12
> **Response to nzQK cont'd**
>
> **The method heavily depends on the pipeline steps, which ILP used, how accurate the oracle assigns**
>
> LogicDP is agnostic to the choice of ILP methods. In the experiments, we evaluated LogicDP with two very different ILP methods, and found the performance difference to be small, indicating LogicDP is also insensitive to the ILP methods. Here, we show the performance on VG and FB15K with three ILP methods to better illustrate this.
>
> | ILP method | FB15K-237 | FB15K-237 |  VG  |
> |:----------:|:---------:|:---------:|:----:|
> |            |    MRR    |  Hits@10  |  R@1 |
> | NeuralLP   |    0.26   |    38.9   | 0.55 |
> | NLIL       |    0.28   |    42.9   | 0.57 |
> | dNL-ILP    |    0.28   |    43.8   | 0.54 |
>
> We have also incorporated the result into the revision and added discussion accordingly.
>
> In terms of the quality of the oracle score, indeed, an inaccurate oracle score will deteriorate the performance, but again, this is also true for any data programming method, for example, with Snorkel, one can write a low-quality labeling function, and thus generate noisy labels.
>
> However, this risk does not undermine this methodology, the key motivation of data programming methods is the human-in-the-loop training paradigm. If an inaccurate score is assigned, humans can readily detect it by inspecting the performance by the end of the iteration and then modifying the score.
>
>
> **The originality is marginal, it is merely a simple application of the data programming**
>
> We respectfully disagree.  While LogicDP shares the same data programming paradigm as prior work such as Snorkel, the specific designs significantly differ and the technical challenges are non-trivial.
>
> Snorkel excels at domains of unstructured data such as text and requires users to manually create labeling functions. Its variant, Snuba, can automate the rule generation for unstructured feature vectors but cannot be readily applied to graph data (Please also see the response to FEgy).
>
> LogicDP, on the other hand, is an automated data programming method for graph data, and to the best of our knowledge, it is the first of this kind. Our framework exploits graph-specific properties and incorporates techniques such as ILP to automate the rule generation process. We also showed that this setting is different from that in the prior work, which calls for special treatments for rule refinement and label aggregation. The proposed components such as the budget-aware function refinement and posterior regularization are also not present in the prior work.
>
>
>
> **No source code or data is provided**
>
> The source code is provided in the draft on page 7 (https://www.dropbox.com/s/1dhvv7og32w2qvd/code.zip?dl=0). All benchmark datasets are publicly available and the pre-processing code is included in the repo.

---

### Official Review · Reviewer_M5YG · 2022-10-26

**Confidence:** 3
**Correctness:** 3
**Technical Novelty And Significance:** 3
**Empirical Novelty And Significance:** 3
**Recommendation:** 8

**Clarity, Quality, Novelty And Reproducibility:**

The paper is very cleat (ok, examples would help)

`That suggest high quality,

Most of the woks in ,ethods seem tobuild open prior work, but the key  is novel (to the best of knwldge).

Repro: ok

**Strength And Weaknesses:**

The main strength of the paper is uding iLP in an unusual directiom.
The factt that the authord use sampling of the original datasets raises the question of scalability.
I felt it wss sometimes a bit unclear what was ols and what was new.

**Summary Of The Paper:**


This paper introduces an ILP based method for adding labels tp unlabelled graph The ussfdulnesss of rge algorithm is supported by an extensive set of experiments,SS



**Summary Of The Review:**

I really liked the basic idea of the paper. Unlabeled data is a big problem,and it is nice to see progress in the area/
The experimentas are welll designod;  I usually feel a bit iffy about human input, but I suppose there is a real need here.

I am not focused on graph learning, so I may ne missing smething. but in general I liked the paperr.

---

> ### Author Response · Authors · 2022-11-12
> **Response to M5YG**
>
> We thank the reviewer for the comments. Our responses are as follows:
>
> **The need for human input**
>
> Indeed, the traditional supervised learning paradigm does not rely on human input. This work falls into the category of the human-in-the-loop paradigm, which seeks to efficiently incorporate human weak supervision into the learning process. The idea that one can convert rich human prior knowledge into noise labels has become the foundation of many data programming and weakly-supervised method.

---

### Decision · Program_Chairs · 2023-01-20

**Decision:**

Accept: poster

**Justification For Why Not Higher Score:**

Not the broadest reviewer support.

**Justification For Why Not Lower Score:**

There are no major flaws identified in the reviews that are not adequately addressed in the author response. What remains is a subjective evaluation of how important the style of work is.

**Metareview: Summary, Strengths And Weaknesses:**

The paper proposes data programming for graph data with ILP-learned labeling rules.
Any human in the loop evaluation is difficult (can humans do this, how much information is provided, how to make baselines equal, etc.) but since this is not an HCI conference, this paper does an adequate and fair job.
The main strength of the paper is the proposed methodology which is interesting and fresh. The proposal is significantly different from classic data programming, and leverages core strengths of ILP.
The meta-reviewer agrees with the authors that interacting through rules is the main strength, rather than a weakness of the proposed method. In general the meta-reviewer found the author response to be very strong.

**Note From Pc:**

if the above contains the word "oral" or "spotlight" please see: "oral" presentation means -> notable-top-5% and "spotlight" means -> notable-top-25%. As stated in our emails, we are disassociating presentation type from AC recommendations